



# Online treatment of eruption dynamics improves the volcanic ash and SO₂ dispersion forecast: case of the Raikoke 2019 eruption

Julia Bruckert[1], Gholam Ali Hoshyaripour[1], Ákos Horváth[2], Lukas O. Muser[1], Fred J. Prata[3], Corinna Hoose[1], and Bernhard Vogel[1]

[1]Institute of Meteorology and Climate Research, Karlsruhe Institute of Technology (KIT), Karlsruhe, Germany
[2]Meteorological Institute, University of Hamburg, Germany
[3]AIRES Pty. Ltd., Mt Eliza, Victoria, Australia

**Correspondence:** Julia Bruckert (julia.bruckert@kit.edu)

**Abstract.** In June 2019, the Raikoke volcano, Kuril Islands, emitted 0.4-1.8 x $10^9$ kg of very fine ash and 1-2 x $10^9$ kg of SO₂ up to 14 km into the atmosphere. The eruption was characterized by several phases or puffs of different duration and eruption heights. Resolving such complex eruption dynamics is required for precise volcanic plume dispersion forecasts. To address this issue, we coupled the atmospheric model system ICON-ART (ICOsahedral Nonhydrostatic – Aerosols and Reactive Trace

gases) with the 1-D plume model FPlume to calculate the eruption source parameters (ESPs) online. The main inputs are the plume heights for the different eruption phases that are geometrically derived from satellite data. An empirical relationship is used to derive the amount of very fine ash (particles <32$\mu$m), which is relevant for long range transport in the atmosphere. On the first day after the onset of the eruption, the modeled ash loading agrees very well with the ash loading estimated from AHI (Advanced Himawari Imager) observations due to the resolution of the eruption phases and the online treatment of the ESPs.

In later hours, aerosol dynamical processes (nucleation, condensation, coagulation) explain the loss of ash in the atmosphere in agreement with the observations. However, a direct comparison is partly hampered by water and ice clouds overlapping the ash cloud in the observations. We compared 6-hourly means of model and AHI data with respect to the structure, amplitude, and location (SAL-method) to further validate the simulated dispersion of SO₂ and ash. In the beginning, the structure and amplitude values differed largely because the dense ash cloud leads to an underestimation of the SO₂ amount in the satellite

data. On the second and third day, the SAL values are close to zero for all parameters indicating a very good agreement of model and observations. Furthermore, we found a separation of the ash and SO₂ plume after one day due to particle sedimentation, chemistry, and aerosol-radiation interaction.

The results confirm that coupling the atmospheric model system and plume model enables detailed treatment of the plume dynamics (phases and ESPs) and leads to significant improvement of the ash and SO₂ dispersion forecast. This approach can

benefit the operational forecast of ash and SO₂ especially in case of complex and non-continuous volcanic eruptions like the Raikoke 2019.



## 1   Introduction

Explosive volcanic eruptions inject particulate matter and gases into the atmosphere, which are then dispersed by atmospheric
transport processes. Volcanic ash can remain airborne for up to a few months (e.g., Jensen et al., 2018; Niemeier et al., 2009;
Robock, 2000) and drift away several thousand kilometers from the eruption point when emitted into the lower stratosphere or
higher. Moreover, oxidation of volcanogenic $SO_2$ leads to formation of sulfate particles as secondary volcanic aerosols. These
aerosols enhance the scattering of sunlight in the atmosphere and, thus, reduce the incoming shortwave radiation reaching the
ground (e.g., Robock, 2000). Therefore, while the dispersion of ash particles mainly matters for aviation safety (Casadevall,
1994), regional public health (Horwell and Baxter, 2006), and local environment and infrastructure (e.g., Stewart et al., 2006;
Spence et al., 2005; Wardman et al., 2012), the fate of $SO_2$ is crucial for predicting the impacts of volcanism on weather and
climate (Robock, 2000; Mather, 2008; Timmreck, 2012; von Savigny et al., 2020).

Forecasting the dispersion of volcanic aerosols in the atmosphere relies on the representation of both the source and sink
parameters and processes. It has been shown that eruption source parameters (ESP) such as mass eruption rate (MER), plume
height, emission profile, and the duration of the eruption can strongly influence the quality of the forecast of the spatial
distribution of the volcanogenic gases and particles (e.g., Harvey et al., 2018; Scollo et al., 2008). The plume height can
be estimated instantaneously by visual, radar, and lidar based or satellite observations. Until a few hours after the onset of
a volcanic eruption when such plume height observations become available, MER usually remains uncertain. Estimates of
the MER include empirical parametrizations based on plume height (e.g., Mastin et al., 2009) partially corrected by wind
effects (e.g., Degruyter and Bonadonna, 2012; Woodhouse et al., 2013) or are derived with 1D plume models (e.g., Folch
et al., 2016). Further uncertainties arise from the choice of the eruption profile (e.g., De Leeuw et al., 2020), i.e., the vertical
distribution of mass. Different approaches exist to parameterize the emission profile, e.g., idealized profiles (Stuefer et al.,
2012), plume-theory-based profiles (Marti et al., 2017), Gaussian-shaped profiles derived from backward trajectory modeling
(Rieger et al., 2015), constant profiles (e.g., Muser et al., 2020; Beckett et al., 2020), or more complex ones derived from the
observations (e.g., De Leeuw et al., 2020). However, all parametrizations of the ESP have in commom that the volcanic cloud
dispersion remains decoupled from unresolved volcanic eruption dynamics including also the influence of the atmosphere on
the emission height. This accounts for large uncertainties in modeling studies at regional to global scales (Textor et al., 2005;
Timmreck, 2012; von Savigny et al., 2020). Marti et al. (2017) overcame this issue by coupling the NMMB-MONARCH-ASH
transport model (Nonhydrostatic Multiscale Model on the B-grid – Multiscale Online Nonhydrostatic AtmospheRe CHemistry
model – ASH) with the 1D plume model FPlume, which calculates the MER and the mass distribution in the column online.
They described the gravitational spreading of the umbrella cloud by the model of Costa et al. (2013). Collini et al. (2013)
highlighted a good agreement in ash transport simulations with satellite observations for the Cordon Caulle eruption 2011 by a
combined WRF/ARW-FALL3D forecast system. Plu et al. (2021) simulated the 2010 Eyjafjallajökull eruption with MOCAGE
model (Modèle de Chimie Atmosphérique de Grande Echelle) and hourly changing MER from FPlume. They highlighted
more concentrated ash concentrations in the horizontal and vertical scale, which more realistically represents the horizontal
dispersion compared to parameterized MERs.





Only ash particles smaller than 32 $\mu$m (hereafter referred to as very fine ash) are relevant for long range transport in the atmosphere (Rose and Durant, 2009). However, the amount of very fine ash emitted by a volcanic eruption is uncertain and depends on different parameters such as the strength and height of an eruption (Gouhier et al., 2019), the composition of magma
(Rose and Durant, 2009), and the availability of water (e.g., van Eaton et al., 2012; Prata et al., 2017). Gouhier et al. (2019) analyzed data of past volcanic eruptions with respect to the fraction of very fine ash in the whole mass erupted. They found that strong volcanic eruptions are less efficient in emitting very fine ash into the atmosphere possibly due to higher sedimentation within the plume. Most forecast models assume a fixed value for the fraction of very fine ash between 1% (e.g., Muser et al., 2020) and 5% (e.g., Webster et al., 2012; Beckett et al., 2020) regardless of the strength of the eruption and lava composition.
Once emitted into the atmosphere, aerosol dynamics (including aggregation) lead to a faster growth of particles and, thus, a quicker removal from the atmosphere (e.g., Brown et al., 2012, and references therein). Muser et al. (2020) investigated the impacts of aerosol dynamics and radiation interactions on the ash dispersion after the Raikoke eruption in June 2019. They showed that aerosol dynamical processes such as nucleation, condensation, and coagulation enhance the removal of the ash particles from the atmosphere. On the other hand, the absorption of incoming shortwave radiation by internally mixed aerosols
leads to the lofting of the aerosol plume. The simulated effect for the Raikoke eruption is in the order of 6 km during the first 4 days (Muser et al., 2020). Zhu et al. (2020) confirmed that coagulation of mixed particles with ash and sulfate is required to produce the evolution of the size distribution of mixed particles following the Kelud eruption 2014. However, they further found that the initial $SO_2$ lifetime is determined by direct $SO_2$ uptake on ash, rather than its oxidation by OH.

Here, we aim to link complex eruption source parameters during the first hours of volcanic eruptions to the fate of vol-
canogenic gases and aerosols. As a case study, we investigate the ash and $SO_2$ dispersion of the Raikoke eruption (48.29°N, 153.24°E) on 21 and 22 June, 2019 during the first three days after the eruption onset. The eruption was characterized by ten 'puffs' of 5 to 14 km height lasting between 5 min and 3 h (Horváth et al., 2021b), which leads to further difficulties in deriving reasonable ESPs for plume dispersion forecasts. The constant emission rate and profile used by Muser et al. (2020) caused an approximately 6 h time lag between the time series of modeled and observed ash mass loading. This gap might be
filled by improving the representation of the eruption source parameters and a varying fine ash fraction according the relationship by Gouhier et al. (2019). Moreover, the impacts of source and sink processes on the fate of $SO_2$ erupted from Raikoke remain unexplored in the study by Muser et al. (2020). Kloss et al. (2021) investigate $SO_2$ transport following the Raikoke 2019 eruption with observations and models. They found enhanced stratospheric aerosol optical depths in the whole northern hemisphere for more than one year following the Raikoke eruption when using an $SO_2$ setup which realistically represents the
transport of volcanic compounds during the first hours after the Raikoke eruption. De Leeuw et al. (2020) found that simulating the correct burden of $SO_2$ is sensitive to the fraction emitted into the lower stratosphere and therefore depends strongly on the emission profile chosen. In this work, we want to answer the following research questions: How large is the influence of resolving the eruption phases on the predicted ash mass loading after the Raikoke eruption? Can an online treatment of volcanic ESPs improve the predicted mass loading and dispersion of ash and $SO_2$ plumes? And what is the impact of aerosol–radiation
interaction on the dispersion of the $SO_2$ plume? The paper is structured as follows: in Sect. 2, the methodology including the model setup, the inputs, observations, and the validation method used are described. In Sect. 3, we evaluate our experiments





with respect to mass loading and structure, amplitude, and location of the plume. In addition, we discuss the separation of the ash and $SO_2$ plumes due to aerosol-radiation interaction. Finally, Sect. 4 concludes the paper.

## 2 Methods

### 2.1 ICON-ART modeling system

In this study, we performed simulations with the global weather and climate model ICON (ICOsahedral Nonhydrostatic model) together with the module for Aerosol and Reactive Trace gases (ART). ICON solves the full three-dimensional non-hydrostatic and compressible Navier-Stokes equations on an icosahedral grid and allows seamless predictions from local to global scales (Zängl et al. (2015); Heinze et al. (2017); Giorgetta et al. (2018)).

ART, being part of ICON, supplements the model by including emissions, transport, gas phase chemistry, and aerosol dynamics in the troposphere and stratosphere (Rieger et al., 2015; Schröter et al., 2018; Weimer et al., 2017). Muser et al. (2020) reported and demonstrated latest improvements in ICON-ART with respect to the AEROsol DYNamics module (AERODYN), which is also used in the present paper. In AERODYN, aerosols are organized in 7 log normal distributions considering Aitken (as soluble), accumulation (as soluble, insoluble and mixed), coarse (as insoluble and mixed) and a giant mode (as insoluble). For each mode, the prognostic equations for number density and mass concentration are solved keeping the standard deviations of the modes constant. For the Aitken mode, nucleation, condensation, and coagulation are considered, while accumulation mode and coarse mode are affected by condensation and coagulation only. Shifting of particles into another mode occurs either when a threshold diameter is exceeded (shift into larger mode) or when a mass threshold of soluble coating on insoluble particles is exceeded (shift from insoluble to mixed mode) (Muser et al., 2020). In AERODYN, water and sulfate (also ammonium and nitrate) can condense on ash particles and therefore change the physical properties of ash, e.g., size, density, optical properties. Changes of particle optical properties can further feedback on the radiation and atmospheric state. However, effects of aerosol dynamics on atmospheric humidity and clouds is not considered yet.

For detailed descriptions of ICON, ART, and AERODYN, we here refer to the works by Zängl et al. (2015), Rieger et al. (2015) and Schröter et al. (2018), and Muser et al. (2020), respectively.

### 2.2 Coupling ICON-ART with FPlume

For a better estimation of the ESPs, we coupled ICON-ART online with the one-dimensional volcanic plume rise model FPlume (Folch et al., 2016; Macedonio et al., 2016). FPlume solves the equations of the buoyant plume theory (Morton et al., 1956) along the vertical plume axis. It includes processes like ambient air entrainment, plume bending due to wind, particle wet aggregation, energy supply due to water phase changes, particle fallout, and re-entrainment of particles (Folch et al., 2016).

Figure 1 summarizes the procedures performed at every time step in which FPlume is active: First, vertical profiles for wind, temperature, pressure, and humidity simulated with ICON serve as meteorological inputs for FPlume. In the second step, FPlume calculates the plume properties, i.e, total MER in case of a given plume height (as here) or plume height in case of a





given MER. Thirdly, the fraction of very fine ash is determined based on plume height and total MER by using the relationship of Gouhier et al. (2019).

In the last step, ash is emitted into ICON-ART by multiplying the MER of very fine ash with the vertical profile derived from the normalized Suzuki distribution, which is the same as the one used by (Marti et al., 2017):

$$S^*(z) = \frac{\mathrm{E}\left(\left(1 - \frac{z}{H_p}\right)\exp\left(4.0\left(\frac{z}{H_p} - 1\right)\right)\right)^{5.0}}{MAX(S(z))} \tag{1}$$

Here, $S(z)$ describes the vertical emission profile, $E$ is the emission rate of very fine ash, $H_p$ is the plume top height, and $z$ refers to the height in the plume. We completely disregarded the mass of particles larger than 32 $\mu$m as this fraction has been

shown to be irrelevant for long-range transport Rose and Durant (2009).

More details on the initialization of the ash particles is given in the next section. Besides ash, we also emitted SO$_2$. Different from the ash emission, we prescribed the MER of SO$_2$ based on satellite estimates, but we released it into ICON-ART with the same profile and phases as the ash. This simplification was necessary as no further information on temporally SO$_2$ emission is available. Yet, during volcanic eruptions in general it is possible that the ash and SO$_2$ are emitted at different phases of the

eruption.

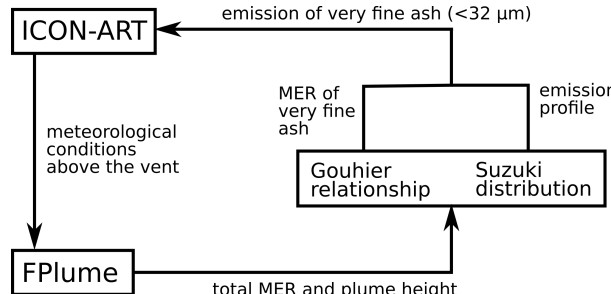

**Figure 1.** Schematic of the setup with the coupling between the global ICON-ART model and the 1D-plume rise model FPlume.

Besides meteorological data, FPlume needs estimates of the exit temperature, exit velocity, exit volatile fraction, and plume height to solve for the total MER. Our setup enables to define these parameters for multiple eruption phases meaning that the MER used for ICON-ART depends on the exit conditions for each plume phase and the meteorological conditions. When solving the plume dynamics for the MER knowing the plume height, FPlume performs the calculations iteratively for a range

of possible MERs to reach the plume height wanted (Folch et al., 2016). Unfortunately, for small eruption heights FPlume sometimes fails to determine the corresponding MER. Therefore, we decided to use FPlume only for plume heights larger than 10 km in the case of the Raikoke 2019 eruption to ensure an uninterrupted simulation. For plume heights below 10 km, the Mastin empirical relationship is used.



### 2.3 Eruption source parameters

#### 2.3.1 Vent conditions

Raikoke emits primary basaltic lava, and, therefore, we assumed the following exit conditions for FPlume: The exit temperature of 1273 K and exit water mass fraction of 3% were the same for all eruption phases (Mastin, 2007); the exit velocity was linearly changed depending on the plume top height relative to 14 km, where the exit velocity was set to 120 m/s. Our resulting MERs are insensitive to the input values in the range of 10%.

The equation for the fine ash fraction by Gouhier et al. (2019) depends on whether the $SiO_2$ content is high or low and whether the conduit was opened or closed. As no information on the conduit has been available so far, we averaged the fine ash fraction for low $SiO_2$ – closed conduit and low $SiO_2$ – opened conduit.

#### 2.3.2 Geometric plume heights

In addition to temperature and exit velocity, FPlume requires the plume height as input to calculate the MER. The height above the ellipsoid of the individual puffs was estimated by a recently developed geometric technique (Horváth et al., 2021a), which exploits the near-limb views provided by Geostationary Operational Environmental Satellite-17 (GOES-17). Such oblique observations offer close to orthogonal side views of vertical columns protruding from the Earth ellipsoid and thereby facilitate a simple height-by-angle method to derive point estimates of eruption column height in the vicinity of the vent. The GOES-17 side view heights were in good agreement with independent geometric estimates derived from plume shadows and GOES-17– Himawari-8 stereo observations (Horváth et al., 2021b).

The Raikoke plume heights cannot be unambiguously determined by the traditional infrared brightness temperature method. For most puffs, the minimum 11 $\mu$m brightness temperature ($BT_{11}$) falls within the narrow temperature range of the quasi-isothermal layer above the tropopause and leads to multiple height solutions within a wide altitude range of 10-24 km. At certain times (e.g., 23:50 UTC on 21 June or 01:20 UTC on 22 June), the massive eruption plume is undercooled, even precluding the application of the temperature method. For the smaller plumes produced by less energetic puffs, on the other hand, the $BT_{11}$ has a warm bias due to contributions from the warmer lower-level marine Sc layer around the volcano, resulting in underestimated heights. A detailed analysis of the Raikoke plumes, including a comparison of the various height estimates, is given in Horváth et al. (2021b). The uncertainty of the plume heights lays within a range of ±500 m.

### 2.4 Model configuration

We performed global simulations with ICON-ART using a horizontal grid size of roughly 13.2 km (R3B07 grid) and 90 vertical levels up to 75 km. For each experiment, we simulated 72 h starting from June 21, 12.00 UTC with initialized analysis data provided by the German Meteorological Service (DWD). During active eruption periods, the ESPs of Raikoke are calculated online with FPlume.





The Raikoke eruption 2019 was characterized by 9 puffs between 18 UTC on June 21 and 7 UTC on June 22 and one

more or less continuous eruption phase between 22.40 UTC and 01.55 UTC. We performed three experiments: 1) a reference experiment in which FPlume calculates ESPs for plume heights above 10 km (below 10 km with Mastin et al. (2009), see section 2.2) with a varying fine ash fraction and aerosol-radiation interaction activated in ICON-ART (FPlume-rad); 2) the second experiment calculates the ESPs same as above, but neglects the interaction of aerosols and radiation (FPlume-norad). The comparison of FPlume-rad and FPlume-norad allows to quantify the lifting of the volcanic plume due to radiation; 3) the

third experiment derives ESPs with the empirical relationship by Mastin et al. (2009), and it emits volcanic compounds with a prescribed fine ash fraction from the reference case (mean value for each phase) along a Suzuki profile (i.e., Eq. 1). It further assumes aerosol-radiation interaction (Mastin-rad). Table 1 summarizes the input parameters for the FPlume-rad experiment associated with the different eruption phases including the input parameters. The time limits for the phases and plume heights are based on satellite images from GOES-17, which are described in Sect. 2.3.2. Due to the 10-minute temporal resolution of

the GOES-17 data, the start and end time of each individual puff is smaller than $\pm 5$ min.

**Table 1.** Model set up and parameters for the individual eruption phases (FPlume-rad).

|          | Time            | Plume height in m | Method | Exit temperature in K | Exit velocity in m s$^{-1}$ | Exit volatile fraction in % | SO$_2$ mass emission rate in kg s$^{-1}$ |
|----------|-----------------|-------------------|--------|-----------------------|-----------------------------|-----------------------------|-------------------------------------------|
| Phase 01 | 17.55-18.20 UTC | 7250              | Mastin | /                     | /                           | /                           | 45320                                     |
| Phase 02 | 18.50-19.05 UTC | 9250              | Mastin | /                     | /                           | /                           | 57811                                     |
| Phase 03 | 19.40-20.05 UTC | 10250             | FPlume | 1273                  | 90                          | 3                           | 64073                                     |
| Phase 04 | 20.40-20.50 UTC | 9250              | Mastin | /                     | /                           | /                           | 57817                                     |
| Phase 05 | 21.20-21.25 UTC | 10250             | FPlume | 1273                  | 90                          | 3                           | 64067                                     |
| Phase 06 | 22.00-22.05 UTC | 11750             | FPlume | 1273                  | 95                          | 3                           | 73467                                     |
| Phase 07 | 22.40-01.55 UTC | 13750             | FPlume | 1273                  | 105                         | 3                           | 85946                                     |
| Phase 08 | 03.40-04.05 UTC | 11750             | FPlume | 1273                  | 95                          | 3                           | 73447                                     |
| Phase 09 | 05.40-05.55 UTC | 13250             | FPlume | 1273                  | 105                         | 3                           | 82822                                     |
| Phase 10 | 07.00-07.10 UTC | 4750              | Mastin | /                     | /                           | /                           | 29683                                     |

Figure 2 shows the plume height and MER of very fine ash that is released into ICON-ART by FPlume (red dots) and the Mastin relationship (blue dots). The fraction of very fine ash relative to the total MER predicted by FPlume is in the order of 1.5 - 3 % (not shown). In most phases, the MER calculated with FPlume is lower than the MER calculated with the Mastin equation, and the difference tends to be higher for larger plume heights. Since the exit parameters are fixed during each phase

in the reference case, variation of the MER derived by FPlume must be due to changes in the atmospheric conditions. As the relationship by Mastin et al. (2009) neglects atmospheric conditions and the fine ash fraction is fixed within one phase, the MERs of the very fine ash are constant within each phase. According to these MER values, the total mass of very fine ash emitted in the model for all puffs together is about $1.37 \cdot 10^9$ kg using FPlume and $1.75 \cdot 10^9$ kg using Mastin-derived MERs.

The total amount of very fine ash is evenly distributed as insoluble tracer over the accumulation, coarse, and giant modes.

Muser et al. (2020) explained the detailed size distribution parameters of these modes.





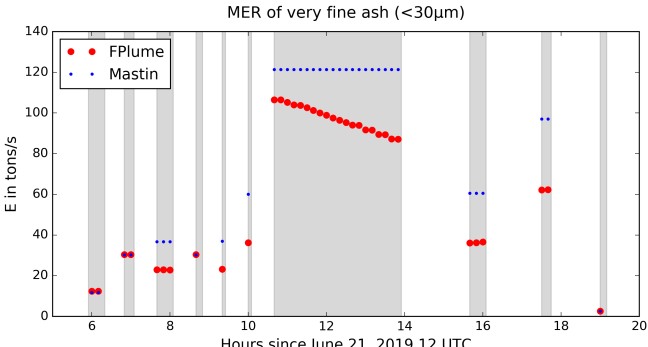

**Figure 2.** Mass eruption rate for very fine ash in tons/s calculated with FPlume times the fine ash fraction from Gouhier et al. (2019) (red) and calculated with Mastin times the fine ash fraction derived in the FPlume experiment (blue). The fine ash fraction is the same in both experiments to allow a direct comparison of the FPlume and Mastin derived MER on the dispersion in the atmosphere. Active eruption phases are indicated by the gray shading.

Following previous studies, we emitted a total of $1.5 \cdot 10^9$ kg SO$_2$ (Muser et al., 2020; Kloss et al., 2021; De Leeuw et al., 2020). However, the SO$_2$ release is linearly adjusted to the eruption heights and length of each phase (Table 1) as follows:

$$E_{ph} = \overline{E}_{SO2} \cdot \frac{H_{ph}}{H_T} \qquad (2)$$

$E_{ph}$ is the phase-dependent MER of SO$_2$, $\overline{E}_{SO2}$ is the mean MER based on the observed amount of SO$_2$ and the sum of the
duration of all phases, $H_{ph}$ is the phase plume height, and $H_T$ is the phase duration-weighted mean plume height. Finally, SO$_2$ release is vertically distributed according to the Suzuki profiles (comparison to previously used profiles in Fig. A1). Here, ICON-ART treats SO$_2$ as a chemical tracer that can be oxidized by a simplified OH-chemistry scheme as presented in Weimer et al. (2017).

### 2.5   Himawari-8 Ash and SO$_2$

To validate the model results, we used column SO$_2$ and ash mass loadings estimated from the 16-band visible and infrared Advanced Himawari Imager (AHI) onboard the Himawari-8 geostationary satellite at every full hour. Himawari-8 is operated by the Japanese Space Agency (JAXA) and the Japanese Meteorological Agency (JMA). A detailed description of the data product and methods used here is already given in Muser et al. (2020) and references therein. In short, SO$_2$ is retrieved by the AHI band centered near 7.3 $\mu$m, where the absorption of SO$_2$ is high. A further retrieval scheme, as described in Prata
et al. (2004), was applied to minimize the interference with vapor and clouds. For volcanic ash retrievals, the AHI bands near 11.2 $\mu$m and 12.4 $\mu$m are considered. The ash retrievals were corrected by a mask that accounts for the effect of pixel containing meteorological clouds but were detected as completely cloud covered. Hereby, only pixels inside a 0.1 g m$^{-2}$ contour line are considered and a 9x9 median filter smooths out 'spikes'.





## 2.6 SAL Method

The SAL method is an object-based quality measure originally developed to verify precipitation forecasts (Wernli et al., 2008, 2009). However, it has also been successfully applied for transport forecasts of volcanic compounds (e.g., Muser et al., 2020; De Leeuw et al., 2020) and was used in this study for volcanic ash and $SO_2$ as well. The method assesses predefined objects based on a threshold value. These objects are then evaluated according to their structure (S), amplitude (A), and location (L). The structure component S compares model and observations with respect to the volume of the defined objects. The

value ranges between -2 and 2. Positive values indicate too large and/or too flat objects, whereas negative values indicate too small and/or too peaked objects. A value of 0 refers to a perfect forecast with respect to the structure. The amplitude component A evaluates the domain-averaged relative deviation of the forecasts from observations and is positive when the model overestimates the predicted quantity and vice versa (it also ranges between -2 and 2). For a perfect forecast of the amplitude, A is zero. The location component L consists of two parts: L1 describes the agreement between forecast and

observation in terms of the normalized difference between the centers of mass, whereas L2 refers to the average distance between the center of mass of all objects and the individual objects. Each of L1 and L2 can reach values between 0 and 1 so that L in total can have values between 0 and 2 with a perfect forecast with respect to the location at L=0 (Wernli et al., 2008, 2009).

For SAL comparison of Himawari-8 and ICON-ART data, we derived 6-hour averages and interpolated ash and $SO_2$ values
onto a regular grid between 120° W and 80° E and 20° N and 85° N with a resolution of 0.1°. However, before interpolation, we applied a 5x5 pixel mean averaging to fill gaps in the satellite data whereby only values different from zero are considered. Otherwise, the linear interpolation would have led to a loss of information when mapping on a coarser grid, because the regular grid is about 4 to 5 times coarser than the retrieval grid.

## 3 Results and discussions

### 3.1 Validation of mass loading

The Raikoke eruption 2019 injected ash and $SO_2$ up to 14 km into the atmosphere. Figure 3 shows mean ash (left) and $SO_2$ (right) column on June 22 (top row) and 23 (bottom row) in our reference simulation FPlume-rad. The volcanic plume first spreads with westerly winds and is then dragged into a low pressure system over the Northern Pacific Ocean. In the mass loadings of both compounds, no clear horizontal separation of the ash and $SO_2$ plume is visible (compare Fig. 3 left and right

side). However, we will further investigate the separation of ash and $SO_2$ due to radiation in Sect. 3.3 after we validated our setup.

Figure 4a shows the temporal evolution of the ash loading in the atmosphere following the Raikoke eruption for different experiments and observations. The Himawari-8 data reveals a steep increase of ash mass at 22 UTC on June 21 until a plateau of 1.0 – 1.1 Tg is reached at 3 UTC on June 22 followed by a descent to 0.3–0.5 Tg after 7 UTC (June 22). Muser et al.

(2020) (green curve) emphasized that aerosol processes account for the ash removal. Nucleation, condensation, and coagulation

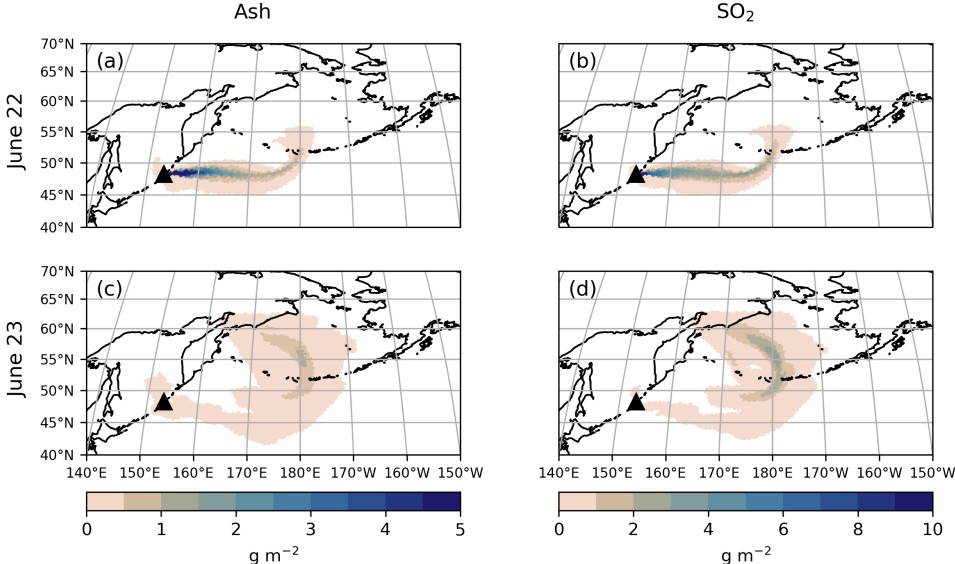

**Figure 3.** Simulated daily mean column mass loadings for ash (left) and SO$_2$ (right) on June 22, 2019 (top row) and June 23, 2019 (bottom row) in g m$^{-2}$. The results are based on the FPlume-rad experiment.

increase the size of aerosol particles and, thus, lead to a faster sedimentation. However, Muser et al. (2020) were not able to quantitatively explain the time lag between model and observations during the first hours of the eruption (6 UTC on June 21 until 3 UTC on June 22). Besides, the continuous emission with a constant MER led to a slight overestimation of the ash mass loading (Muser et al., 2020). We have closed these gaps as follows.

The maximum of total ash derived with ICON-ART coupled with FPlume in both experiments with and without radiation-aerosol interaction coincides very well with the Himawari-8 data (compare red and yellow curve with black curve). The total ash derived with Mastin (different MER but the same fine ash fractions and emission profile as in the FPlume experiments) overestimates the amount of ash during the first 12 h after the onset of the eruption (blue curve). Thus, neglecting meteorological effects and other plume-related processes in the case of the Raikoke eruption results in higher MER in the long continuous

phase of the eruption and subsequently to an overestimation of the ash emissions into ICON-ART (Fig. 2).

All simulation experiments in Fig. 4a include aerosol dynamics and have correctly reproduced the fallout of particles as indicated by the decrease of ash after two days. From Fig. 4b, where the temporal development of the different modes is shown, we can conclude that the decrease of ash after two days is mainly due to coarse and giant mode particles. The total ash from the simulations with FPlume display the best agreement with the Himawari-8 data in this analysis (Fig. 4a). However,

the other curves remain mostly within the error range of Himawari-8 data as well (gray shading). Thus, we conclude that the online treatment of plume development improves the ash loading prediction during the first hours and days after the eruption. Afterwards, the aerosol dynamical processes become more important, and the differences between the experiments decrease.





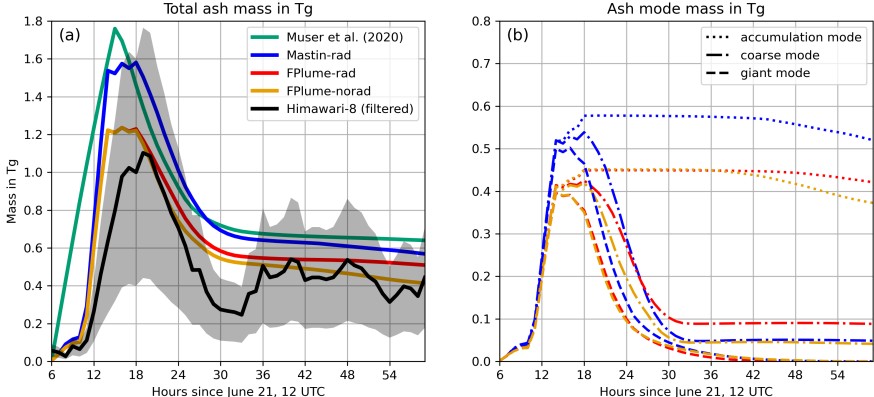

**Figure 4.** (a): Temporal evolution of the total amount of ash in the atmosphere in Tg for Himawari-8 observations (black) with an estimated uncertainty range (gray shading), simulated by Muser et al. (2020) with one constant emission phase (green), simulated with ICON-ART with and without aerosol-radiation interaction, each with phase-depending FPlume MERs, and a fine ash fraction derived with the relationship by Gouhier et al. (2019) (red and yellow), and simulated with ICON-ART, eruption-phase depending Mastin MERs, and the same fine ash fraction as for the red curve (blue). (b): Temporal evolution of the simulated mass for the different ash modes (dotted: accumulation mode; dash-dotted: coarse mode; dashed: giant modes). The colors refer to the model experiment shown in the left panel.

## 3.2 Validation of dispersion using SAL

For the quantitative validation of the forecast quality, we performed a SAL analysis based on the values of 6h averaged $SO_2$

and ash mass loadings. We compare the results of FPlume-rad experiment and Himawari-8 satellite data. Figure 5 shows the values for the structure on the abscissa, the amplitude on the ordinate, and the location in colors.

The location of the $SO_2$ plume agrees very well between model and observations throughout the whole simulation period. This is shown by the location values which are close to zero. The structure and amplitude values are close to zero between 24 to 72 hours after the beginning of the simulation on June 21 at 12 UTC. Thus, there is a high agreement between model and

observations during this period. However, during the first 24 hours, the model prediction shows higher amplitude values and a low structure value, indicating a larger mass loading in the model and a less diffuse $SO_2$ cloud in the model compared to the satellite estimates. We argue that the discrepancy in the amplitude between model and observation during the first hours of the Raikoke eruption stems from the possible underestimation of $SO_2$ by the satellite retrievals due to the dense ash cloud covering the region around the volcano. Moreover, several studies found that including parametrizations for the gravitational

spreading of volcanic plumes in dispersion models is important to reproduce main features of the volcanic eruption cloud (e.g., Costa et al., 2013; Webster et al., 2020). We cannot rule out that the gravitational spreading is inadequately in the beginning of the simulation due to our relatively coarse horizontal resolution of 13 km. This might partly explain the large difference in the structure between ICON-ART and Himawari $SO_2$ during the first hours. In addition, our simulation is also affected by the uncertainties of input parameters (e.g., start and end time of individual puffs, plume heights, exit conditions).





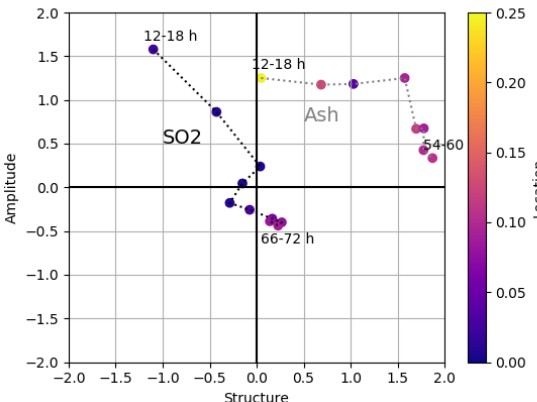

**Figure 5.** SAL values for ash and $SO_2$ in the reference experiment (FPlume-rad): with S(tructure) on the abscissa, A(mplitude) on in the ordinate, and L(ocation) in colors. The dotted lines connect consecutive hours to visualize the temporal development (black for $SO_2$ and gray for ash). The labels of the hours are given relative to the start of the simulation on June 21, 2019 at 12 UTC.

The model also predicts the location of the ash plume very well (Fig. 5). However, the positive amplitude during the first day and positive structure values throughout the whole simulation indicate that the modeled ash loading is too high at the beginning and more diffusive over the domain. Figure A2 (first and second column) compares all 6 h–mean ash loadings. The large spread of the modeled ash plume across large parts of the northern Pacific Ocean is not seen in the observations. We argue that Himawari-8 measurements of ash at this time are might be hampered by water and ice clouds overlapping and obscuring

the ash plume. As for $SO_2$, the S values for ash between 12 and 18 h are slightly negative and, thus, different in sign from the other time steps.

### 3.3   Vertical separation of $SO_2$ and ash plume

In this section, we discuss the evolution of the ash and $SO_2$ plume top heights and focus on the radiative effects on the plume dynamics.

Figure 6 shows the top height of the ash and $SO_2$ plume for the FPlume-rad and FPlume-norad experiment (a), the resulting vertical temperature difference on June 23, 12 UTC (b), and the vertical distribution of $SO_2$ and ash mixing ratios on the same date (c). We picked this particular day, because it allows a direct comparison to Fig. 8 in Muser et al. (2020), which only shows the ash plume top height. Our definition of the plume top height is the same as in Muser et al. (2020): we separated the ash and $SO_2$ plume from the background concentrations by applying a threshold value above which a model grid box is considered as

volcanic plume. The threshold values are 0.01 $\mu$g kg$^{-1}$, 1 $\mu$g kg$^{-1}$, and 10 ppm for the accumulation ash mode, coarse ash modes, and $SO_2$, respectively. The lines for the plume top height and mass averaged height are smoothed by a Savitzky-Golay filter to remove 'steps' due to the low vertical resolution in upper atmospheric model levels.





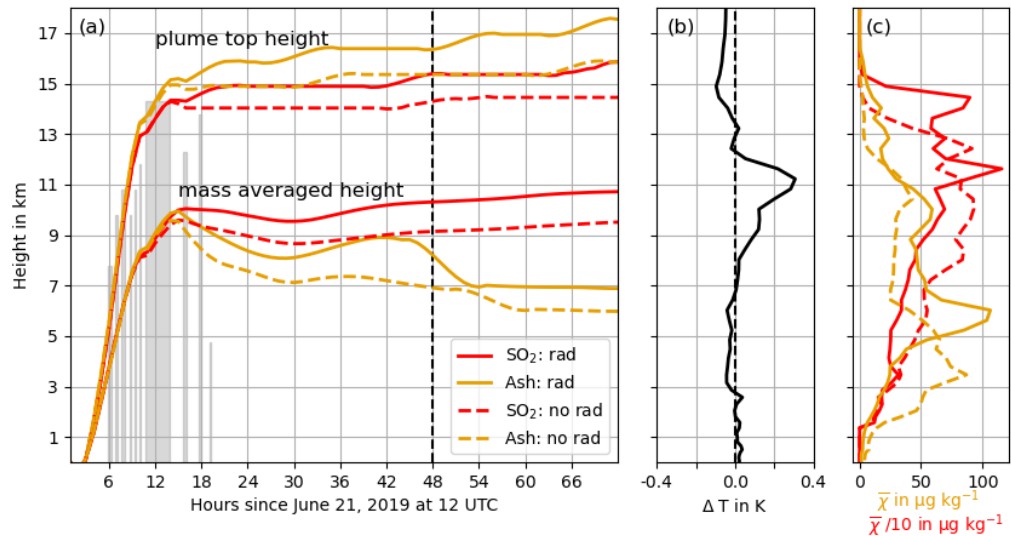

**Figure 6.** (a): Temporal evolution of the $SO_2$ (red) and ash (yellow) plume top height and mass–averaged height for the FPlume-rad (solid) and FPlume-norad (dashed) experiment smoothed by a Savitzky-Golay filter. The grey bars indicate the duration and height of the 10 individual puffs; (b): vertical profile of the temperature difference between the FPlume-rad and FPlume-norad in the ash plume 48 h after the start of the simulation. (c): vertical $SO_2$ (red) and ash (yellow) profile after 48 h for the FPlume-rad (solid) and FPlume-norad (dashed) experiment.

During the first hours after the beginning of the eruption, the plume top height for ash and $SO_2$ mainly rises due to the higher eruption heights of later puffs. The gray bars, which indicate the eruption height, coincide well with the top height (Fig. 6a).

Shortly after the end of the long eruption phase, we clearly see a separation of the ash plume top height between the FPlume-rad and the FPlume-norad experiments. The effect of the ash lofting due to radiation was already investigated in detail by Muser et al. (2020) with the same model system. They found that the absorption of shortwave and longwave radiation by the coated ash particles leads to the warming and rising of the ash plume. We compare the vertical profile of the temperature difference between the FPlume-rad and FPlume-norad case here with the vertical temperature differences in Muser et al. (2020)

on June 23, 12 UTC. A single large positive anomaly of approximately 0.3 K near 11 km occurs in our simulation (Fig. 6b). Subsequently, the profile of the vertical mixing ratio, which in the FPlume-norad case still shows the overlap of the different phase dependent profiles, smooths out in the FPlume-rad case. Additionally, the whole ash plume rises to higher altitudes. In contrast, Muser et al. (2020) found two distinct temperature anomaly peaks around 10 km and 14 km in the order of 0.25 K each, which result in the formation of two maxima in the ash mixing ratios near 10 km and 15 km. The resulting uplift during

the first 12 h in the ash plume is reduced by about 24% in our simulation compared to Muser et al. (2020).

    In the first hours during and after the eruption, the absorption-induced warming of the ash cloud also causes the $SO_2$ plume to rise in FPlume-rad (Fig. 6a and c). However, as $SO_2$ itself absorbs neither solar nor terrestrial radiation in our model setup,

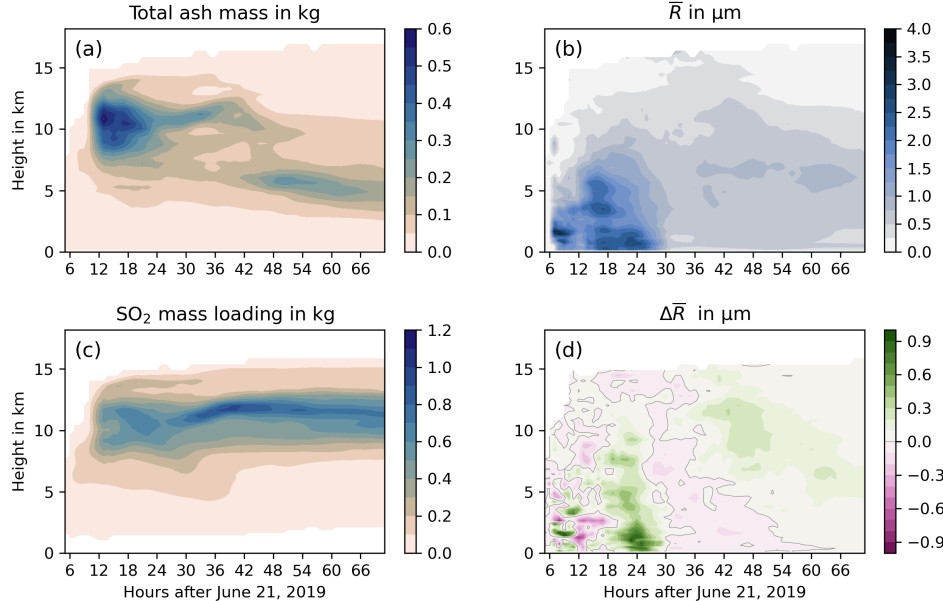

**Figure 7.** Left: Temporal evolution of the horizontally averaged vertical distribution of ash mass (a) and $SO_2$ mass (b) in the FPlume-rad experiment. Right: number-weighted median ash radius (Eq. 3) for FPlume-rad (c) and difference between FPlume-rad and FPlume-norad (d).

the ash plume top height clearly separates from the $SO_2$ plume top height with increasing time (Fig. 6a). The vertical profiles of the $SO_2$ mixing ratio in the FPlume-rad and FPlume-norad case indicate that radiation interaction smooths and reduces the

vertical gradient of the $SO_2$ mixing ratios in the troposphere. In the stratosphere, a second peak occurs above the maximum emission height (Fig. 6c).

The evolution of the mass–averaged height of the ash and $SO_2$ plume indicate an opposite behavior than the plume top height. The mass average of the $SO_2$ plume is generally higher than for the ash plume. In Fig. 7a and c, the vertical distribution of the total ash mass and the total $SO_2$ confirm that the $SO_2$ plume is about 4 km higher on average than the ash plume after

three days. This is in agreement with several existing studies (e.g., Robock, 2000; Timmreck, 2012), which emphasized a fast removal of ash after volcanic eruptions related to the higher weight of ash particles compared to $SO_2$. The step-wise reduction of the ash in the mass averaged height is related to the loss of the giant mode during the first 24 h and the a large fallout of the coarse mode until about 50 h relative to simulation start (Fig. 4b and Fig. A3).





In the next step, we want to compare the vertical distribution of the ash particle median radius $\overline{R}$, which we calculated as follows:

$$\overline{R} = \frac{\sum\limits_{i=1}^{5} r_{m,i} \cdot N_i}{\sum\limits_{i=1}^{5} N_i} \qquad (3)$$

For the overall median radius $\overline{R}$, we consider the five ash modes $i = 1, 5$ (insoluble and mixed accumulation mode, insoluble and mixed coarse mode, and giant mode) and calculated $\overline{R}$ at every grid cell. $r_m$ is the median radius from the log normal distribution and $N$ is the number of particles per grid box. The vertical distribution of the horizontally averaged median radius in Fig. 7 also shows the loss of the larger particles (coarse and giant mode) during the first 24 h. Afterwards, the values of the mean radius are below 1.5 $\mu$m with a maximum around 5 to 6 km. In comparison to the FPlume-norad experiment, the radius is higher on average because aerosol radiation–interaction slows down the removal of larger particles from the atmosphere (Fig. 7d). This effect is especially visible for the removal of the coarse mode which is reduced and delayed of about 2 h in FPlume-rad (maximum around 24 h in Fig. 7d and Fig. 4b). The temporal removal of the giant mode only shows small differences between FPlume-rad and FPlume-norad (Fig. 4b). The larger mean radius after approximately 42 h is related to an increasing removal of accumulation mode particles in FPlume-norad compared with FPlume-rad (Fig. 7d and Fig. 4b). However, we will leave a detailed analysis of the processes changing particle radii to further work.

Both the main $SO_2$ mass and the main ash mass are restricted to a narrower vertical range after three simulation days compared the end of the eruption (around 13 h after simulation start). The location of the $SO_2$ is between 8 and 14 km and between 4 and 10 km for ash (Fig. 7). Thus, for initializing long-range and climate simulations, a release of $SO_2$ and ash at these altitudes is justified if the sedimentation during the first hours is considered in the total emission rate.

Despite of the clear vertical separation of the ash and $SO_2$ plume, the horizontal separation in model and observations is only small in the first three days after the eruption. Nevertheless, a strong vertical wind shear can result in the horizontal separation of the ash and $SO_2$ plume for longer time scales as in Kloss et al. (2021).

## 4 Conclusions

We investigated the Raikoke 2019 eruption, which was characterized by nine puffs and one continuous eruption phase of almost 3-hour-duration. Here, we describe a model setup in which the eruption source parameters were improved by (1) coupling ICON-ART with FPlume to account for the effect of changing volcanic and meteorological conditions and (2) a delineation of eruption phases ('puffs'). We further investigated the effect of radiation-aerosol interaction on the $SO_2$ plume due to a warming of the ash plume. The main outcomes are:

1. We demonstrated a large improvement of the total ash burden forecast in the first 12 h by resolving the individual puffs of the Raikoke eruption, which reduces the ash mass overestimation from 37% to 25%. Additionally, the online calculation

of ESPs by FPlume further improves the simulated ash mass forecast and reduced the overestimation to approximately 6%.

2. In addition to the mass loading, the predicted dispersion of the ash and $SO_2$ cloud also agrees well with observations from Himawari-8 as our SAL analysis revealed. However, the validation of the simulated ash and $SO_2$ dispersion was partially hampered by a dense ash plume in the beginning of the eruption and by overlapping water and ice clouds later on.

3. As already demonstrated in Muser et al. (2020), aerosol radiation interaction leads to a warming of the volcanic ash plume and, therefore, to a lofting of ash particles during the first hours. Here, we additionally found a lofting of the $SO_2$ plume during the first 12 h after the eruption caused by a warming of the plume. However, with increasing time, the $SO_2$ plume becomes more and more vertically separated from the ash plume and the lofting slows down. This is related to a faster sedimentation of the ash particles compared to $SO_2$ and the fact that $SO_2$ does not absorb solar radiation in our 365     model.

*Code and data availability.* The output from ICON-ART simulations performed in this study can be provided upon request by the corresponding author. The ICON-ART code is licence protected and can be accessed by request to the corresponding author. Himawari-8 AHI datasets that have been analyzed in the scope of this study can be provided upon request by the corresponding author.

*Author contributions.* JB, GAH, LOM, CH, BV developed the ICON-ART code and carried out simulations. ÁH provided plume heights
and durations based on GOES-17 data. FJP processed the Himawari-8 data, did the ash and $SO_2$ retrievals. JB and GAH prepared the paper with significant contributions and comments on the original draft from all authors.

*Competing interests.* The authors declare that they have no conflict of interest.

*Acknowledgements.* This research has been funded by the Deutsche Forschungsgemeinschaft (DFG) as part of the Research Unit VolImpact
(FOR2820, DFG Grant No.398006378). The contributions are within the VolImpact sub-projects VolPlume (JB, GAH, BV and ÁH) and VolCloud (CH). JMA/JAXA are acknowledged for providing Himawari-8 data. We thank Fabio Crameri for the development of scientific colour maps (Crameri, 2021) to prevent visual distortion of the data and exclusion of readers with colour-vision defencies (Crameri et al., 2020). Furthermore, we acknowledge support by the KIT Publication Fund of the Karlsruhe Institute of Technology.



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





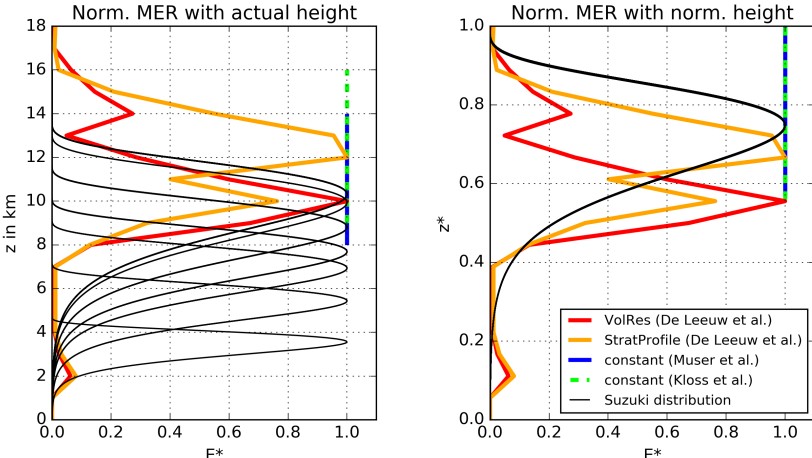

**Figure A1.** Comparison of Suzuki profile used in this work with profiles from previous studies. Left: Normalized with respect to MER, but with actual emission heights; Right: Normalized with respect to MER and emission height.



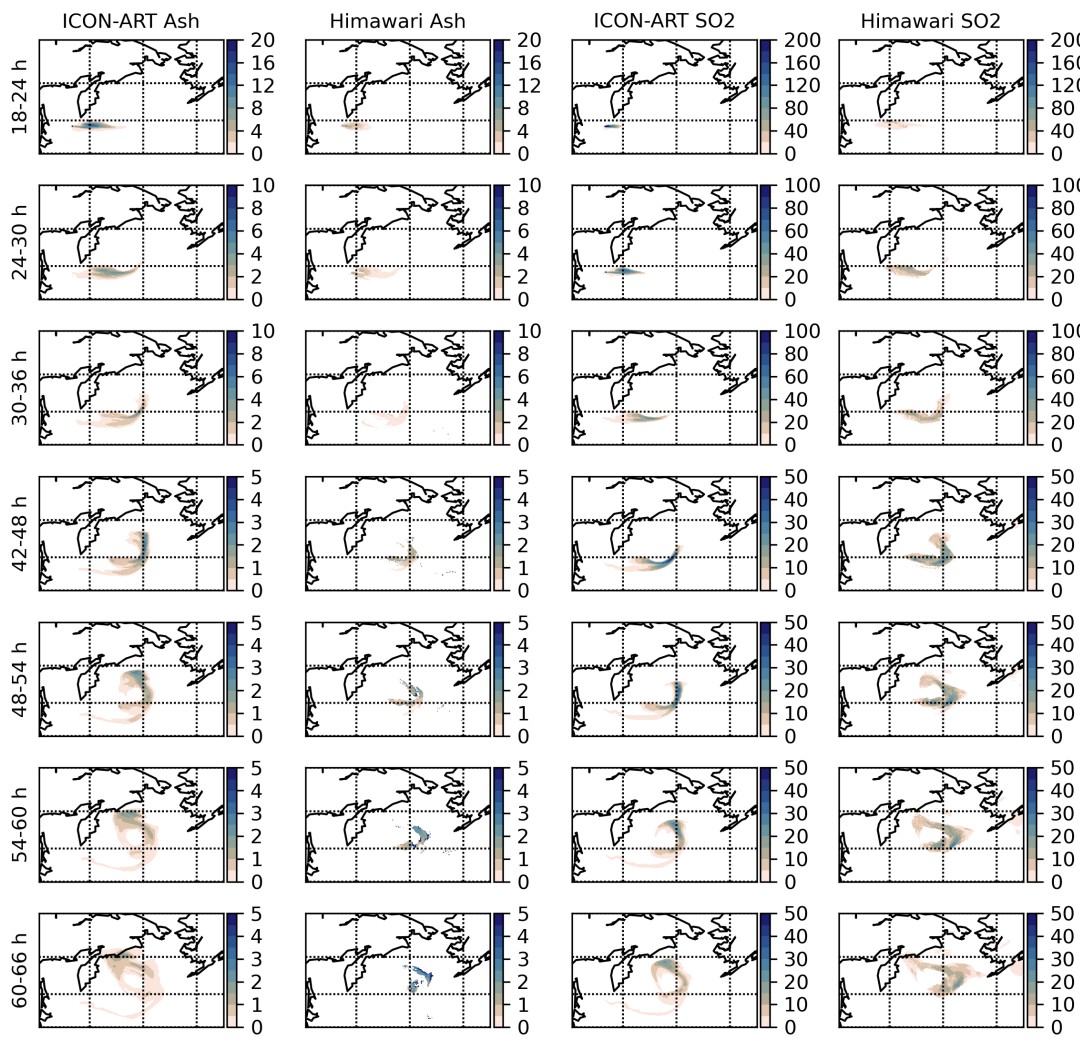

**Figure A2.** Comparison plots for seven ash and SO$_2$ 6-h averaged column loadings (row 1 to 7) in order to explain the discrepancy between simulated ICON-ART data and Himawari-8 observed data in the SAL analysis. First and second column: ash column loadings from ICON-ART and Himawari-8; third and fourth column: SO$_2$ column loadings from ICON-ART and Himawari-8



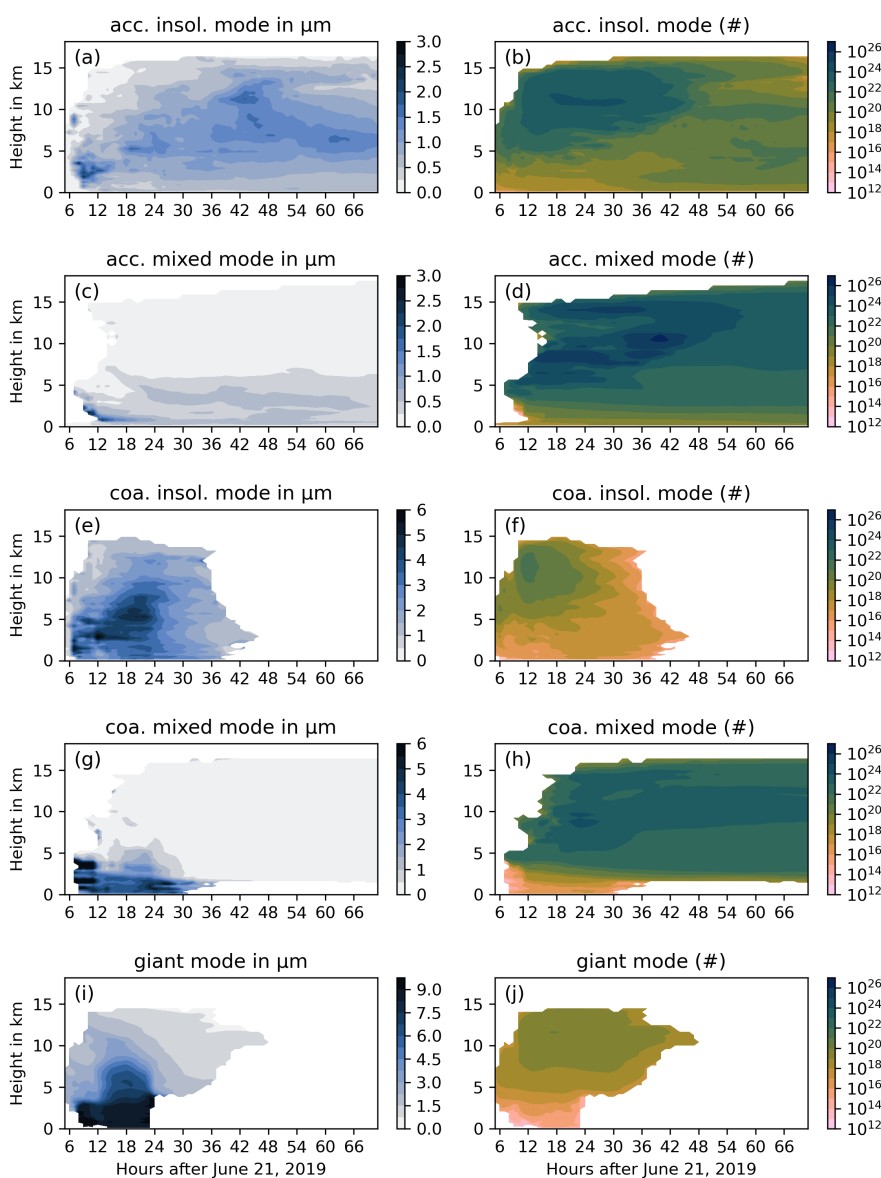

**Figure A3.** Horizontally averaged ash particle median radius of the log normal distribution in $\mu$m (left) and horizontal sum of the particle number (right) of the different ash modes: insoluble accumulation (a, b), mixed accumulation (c, d), insoluble coarse (e, f), mixed coarse (g, h), and giant mode (i, j).