# Peer review of "Online treatment of eruption dynamics improves the volcanic ash and $SO_2$ dispersion forecast: case of the Raikoke 2019 eruption"

_Atmospheric Chemistry and Physics, 2021_

## Author Comment (AC1)

We thank Dr. Folch for the constructive comments and suggestions, which helped us a lot to improve the manuscript.

We decided to re-run our simulations due to different comments in all three reviews. This leads to slightly different results, but does not change the main arguments of the paper. Please find our answers to your comments below (Reviewer's comments in **bold**, our replies in standard font, and modifications of the text in blue).

**General Comments**

- **The FPLUME model actually outputs MER and vertical distribution of mass from plume height. However, it seems that, in the online ICON-FPLUME coupling, only MER is considered whereas a Suzuki parameterisation (with fixed parameter values of 4 and 5) is imposed for the vertical distribution of mass. Is there a particular reason for this? To derive both ESPs (i.e. MER and profile) from FPLUME would seem a much more consistent approach. Note that, in addition, FPLUME gives size-resolved vertical profiles, whereas the authors assume the same values for the Suzuki parameters (4 and 5 according to eq.1) for all particle/aerosol bins.**

  In our work, we only used MER from FPlume and vertically distributed the mass by a Suzuki distribution instead of using the vertical distribution of mass from FPlume. Although, we agree that using the FPlume profiles for mass would be more consistent, we decided on our approach due to two main reasons: 1) based on offline analysis we figured out that the mass profiles for the predefined bin sizes strongly depends on the assumption of initial total grain size distribution (TGSD). As information on the TGSD is often lacking, using FPlume mass profiles lead to a less generic approach and large uncertainties, 2) the definition of ash modes in ICON-ART is only realistic for ash dispersion in the atmosphere and differs from TGSD at the vent. Thus, we would have to convert the FPlume size bins into ICON-ART mode which makes it complicated to control the emission into ICON-ART.

  We would agree that the FPlume profiles might be the ideal solution to study size distributions close to the eruption in test cases. However, for operational purposes where quick responses are needed, considering the FPLUME profiles might not be optimal.

- **Vent conditions (Sec.2.3.1). As noted in the text, FPLUME is sensitive to vent conditions, particularly in the case of low plumes. Insufficient momentum (low exit velocity) and/or insufficiency energy at inlet can**

**yield, together with other atmospheric factors, to lack of model convergence. This is not necessarily a model shortcoming, but actually reflects configurations in which a plume could not be sustained and therefore collapses. Please check if this could be the case for columns below 10 km, particularly if exit velocities drop below 80-90m/s (this seems to be the case according to values from Table 1). Rather than using Mastin-derived MERs (see next comment), I suggest re-running FPLUME for the Raikoke phases 01, 02, 04, and 10 with higher values of exit velocity (see if this converges FPLUME). On the other hand, it is also true that FPLUME is a model for sustained plumes. Application to short-lived burst-like transient puffs may certainly fail.**

Thanks for this comment. We re-run our simulations with higher exit velocities and were able to simulate also the smaller eruption phases online. We note that these modifications affect only the small phases that lead to a negligible effect (<10%) on total mass.

- **I do not see the need for the Mastin-derived modelling option in this study. Which is the purpose of it? First of all, with respect to FPLUME-rad only affects the strength of the source (given that same Suzuki is used in both cases). It should, therefore, imply only a scaling of the Amplitude component of the SAL metric. Unfortunately, comparison (e.g. in Figure 5) is not given to check this. If this is modelling option not relevant to the paper it could be removed. Results and conclusions would be unaltered. The paper could then focus on: i) comparison with previous off-line (Muser 2020) and, ii) comparison rad/norad. This would be simpler and easier to follow.**

We do not show the SAL values for exactly the reason that you mentioned here: The difference in the SAL analysis between FPlume-rad and Mastin-rad is mainly the different Amplitude value for ash. However, we want to show the Mastin-rad experiment here as well because this parametrization is commonly used in volcanic dispersion forecasts. We argue that online treatment of plume dynamics improves the mass of ash in the model. We highlighted our findings in L. 287-288:

Thus, neglecting meteorological effects and other plume-related processes in the case of the Raikoke eruption (offline treatment), as it is often done in volcanic dispersion forecasts, results in higher MER especially in the long continuous phase of the eruption and subsequently increased ash emissions into ICON-ART (Fig. 2).

- **ICON model configuration (section 2.4). Is there any particular reason for running ICON globally? Can the model be run only over a limited area? If not, could you comment on the grid approach over the area of**

**interest or, if none, is the resolution of the R3B07 configuration uniform across the globe**

ICON model can be used in limited area mode (for regional modeling) or with nests (for zooming into particular areas and see scale interactions). We performed global simulation, because 1) it is the same ICON configuration used at German Weather Service for global weather forecast, thus leading to seamless and consistent initialization; 2) it was comparable with the configurations used by Muser et al (2020). Nevertheless, the icosahedral grid of ICON ensures a uniform resolution across the globe (exceptions are nest, but then a refinement of the grid at specific locations is wanted), because the globe is divided in triangles of equal size.
We added the following sentence in the manuscript (L. 184):

The global icosahedral grid of ICON ensures a uniform resolution across the globe.

- **Figure 5. Why it only shows results for FPLUME-Rad? Why the other FPLUME.norad and Mastin-Rad are never shown? On the other hand, can you comment on the poor Structure-component results for ash? Any particular reason for doing 6-hourly model output averages? Considering that Himawari-8 observations can be available 4-times hourly, other approaches having less impact on the results could be considered.**

In this figure we focus on validation of the plume dispersion. We do not show the Mastin-Rad case in the SAL plot, because this experiment only impacts the Amplitude as already answered above. The differences in the SAL values for FPlume-rad and FPlume-norad are only small and a SAL comparison between both experiments does not give additional insights. The impact of radiation on vertical distribution of the plume is discussed in section 3.3.

Our SAL analysis uses modeled and observed data at every full hour as we wrote the output for the 3D-fields only every hour. Therefore, we also used the Himawari data at every full hour. The reason for the 6h averages is mainly that we can get rid of the gaps in the gridded Himawari data. These gaps arise when mapping the data from the native format. Another reason is that we can better compare the results with the maps in Fig. A3.

- **Averaging over such long intervals smooths out (e.g. peaks) and has substantial impact on instantaneous model results. How is SAL affected by this? Can this mask disagreements?**

Averaging smooths out peaks in both observational data and model data. In our analysis, we do not compare long term averages with instantaneous

model results, so the SAL analysis does not compare 'smoothed' observed data with 'raw' modeled data. We attached the results with averaging over 4 and 5 hours for ash here as a comparison (stars: 6h; dots: 4 / 5 h). In the manuscript we still show the SAL analysis with 6 hour averages, because it is easier to compare to Fig. 4 (plotted grid lines agree with dots in SAL plot).

[Figure]

Fig. 1: Ash SAL values for 4-h-averages (black curve with colored dots) and 6-h-averages (grey curve with colored stars).

[Figure]

Fig. 2: Ash SAL values for 5-h-averages (black curve with colored dots) and 6-h-averages (grey curve with colored stars).

**Specific Comments**

- **L36 (and throughout the text). Should multiple reference citations be ordered chronologically?**

We corrected the citations.

- **L103. Could you specify the size ranges for each aerosol mode?**

We added to section 2.4. (L. 216-218)

The three insoluble modes are emitted as lognormal distributions with median diameters of 0.8, 2.98, and 11.35 µm, respectively. The standard deviation is 1.4 for each mode.

- **L166. Sc layer?**

Changed to 'stratocumulus layer' (L. 177).

- **Table 1. Please specify that heights are a.s.l. (and not above vent as required by FPLUME) and that emission rate of SO2 is computed using (2). Also, the emission rate of ash is not reported.**

We specified heights as above sea level and the emission rate of SO2 is computed using Eq. 2. However, this table only shows the input values that are fixed for the individual phases. The emission rate is not fixed within the phases and is therefore shown in Fig. 2.

- **Figure 2. Why is FPLUME (red dots) given in all phases?**

The values refer to both the FPlume-rad and FPlume-norad scenario. We changed the label to FPlume-rad / FPlume-norad to clarify this.

- **Figure 3. Is the color scale adequate? Should the higher value be at around 2 gm-2 or similar?**

We double-checked the plotting script, the color scale is correct. The highest values are selected based on the values appearing in the plots.

---

## Author Comment (AC2)

We thank Dr. Barsotti for the constructive comments and suggestions, which helped us a lot to improve the manuscript.

We decided to re-run our simulations due to different comments in all three reviews. This leads to slightly different results, but does not change the main arguments of the paper. Please find our answers to your comments below (Reviewer's comments in **bold**, our replies in standard font, and modifications of the text in blue).

- **please clarify somewhere in the text if the plume height is intended above sea level or above ground.**

  We clarified plume heights above sea level and above vent where necessary. Additionally, we added in the Methods section (L. 181-182):

  As FPlume requires the plume height above the vent, we converted the GOES-17 above-ellipsoid heights by subtracting a vent height of 550 m.

- **the multiple reference to Muser et al. 2020 paper, makes the reading sometime difficult and incomplete**

  We changed to the abbreviation M20 instead of Muser et al. 2020.

- **I'm worried there is some confusion about the use of the word „plume" which often is also used instead of „cloud". Please clarify throughout the text so that the readability and the understanding of the results will be improved.**

  When we refer to 'plume', we always mean the part originating from the volcanic emission. Thus, it is also the ash and SO2 that is spreading in the atmosphere. We only want to use the word 'cloud' in a meteorological context. However, we double checked our manuscript and had to corrected some sentences in which we used the word 'cloud' instead 'plume', which was indeed inconsistent in the original manuscript.

- **Please be consistent throughout the text if you refer to ash or fine ash.**

  Done.

- **Please in the abstract specify how long lasted the eruption.**

  Done (L. 3):

  The eruption was characterized by several eruption phases of different duration and height summing up to a total eruption length of about 5.5 h.

- **Line 7: what do you mean with „the simulated effect... is in the order of 6 km??"**

  We clarified and rearranged the sentence as follows (L. 70-71):

  „...the simulated lofting effect for the Raikoke eruption resulted in a 6 km rise of the plume top after the first 4 day.

- **Line 103: 7 in word**

  Done (L. 104).

- **Line 126: ... by Marti et al. (2017)**

  Done (L. 127).

- **Line 135: please provide references for this statement.**

  We added L. 139 to the main text and L. 523-524 to the reference section:

  … (e.g., Thomas and Prata, 2011). (L. 139)

  Thomas, H. E. and Prata, A. J.: Sulphur dioxide as a volcanic ash proxy during the April–May 2010 eruption of Eyjafjallajökull Volcano, Iceland, Atmospheric Chemistry and Physics, 11, 6871–6880, https://doi.org/10.5194/acp-11-6871-2011, 2011. (L. 523-524)

- **Line 142: what do you mean with: „... to ensure an uninterrupted simulation?"**

  We decided to re-run the simulation based on different arguments of the reviewers. One argument was to initialize in the FPlume experiments also the eruption phases <10km with FPlume, but with a higher exit velocity (instead of Mastin equation for the MER calculation). Thus, we now could remove the sentence that you commented as unclear.

- **Paragraph 2.3: the Radius of the vent is also part of the equation, so I'm wondering which value or range of values you adopted here?**

  In FPlume, the vent radius is not an input parameter. The following equation relates the MER $M_0$, the vent radius $r_0$, the plume density at the vent $\rho_0$, and the exit velocity $u_0$ (Folch et al., 2016 and FPlume code):

  $$M_0 = \pi\, r_0^{\,2}\, \rho_0 u_0$$

  $M_0$ is first calculated with the method of bisection in case the plume height is given: depending on whether in the loop step before the fixed (given) height is

overestimated or underestimated $M_0$ is either lowered or highered. Afterwards, $r_0$ is calculated with the equation above and further used for the calculations of the plume radius, the entrainment coefficients, and the umbrella height. Thus, we did not adopt values for the vent radius here.

- **Line 166: what Sc layer means?**

Changed to stratocumulus layer in L. 177.

- **Line 176: please refer to Table 1**

Done (L. 190).

- **Table 1: I'd suggest to make this table more complete. I'd add in the first column the day the phases refer to. I'd then also add a column specifying the Fine fraction flux (kg/s) or specify in the caption of the table that the flux for fine ash is shown in Figure 2. Please specify in the caption how the SO2 flux is estimated. Please explain the caption the source of all data showed in the table.**

This table summarizes our input values for the simulation, which are fixed during each individual phase. Therefore, we only added a column with the day, specified in the caption that the flux for the very fine ash is shown in Fig. 2, and included how the SO2 flux is estimated. We cannot provide the very fine ash fraction flux, because this quantity is calculated online and varies between the time steps and not only between phases. We added to the caption of table 1:

The definition of the phases and plume heights above sea level (a.s.l.) are based on GOES-17 satellite observation as described in sect. 2.3.2. The exit conditions are based on typical values of basaltic eruptions as described in Sect. 2.3.1. The SO2 mass emission rate is based on an observational estimate of the total SO2 mass following the 2019 Raikoke eruption from M20, which was distributed over the individual phases with Eq. 3 . This table only shows the values that are predefined and fixed for the individual phases. The temporally varying MER of the very fine ash, which is derived with FPlume and the relationship by (Gouhier et al., 2019), and which is released into ICON-ART, is shown in Fig. 2.

- **Line 186: Figure 2 shows only the MER and not the height**

We corrected it (L. 201):

Fig. 2 shows the MER of very fine ash ...

- **Figure 2: Fine ash is <30 or <32 micron? Please correct the title of the plot. What means E on the y-axis? Please clarify. In the caption: ...calculated with Fplume MER times.....and calculated with Mastin MER times…**

  Done.

- **Line 200: $H_T$ is not the plume height averaged over the entire eruption duration? Please clarify.**

  $H_T$ is a quantity averaged over the entire eruption duration, however longer phases are weighted more strongly. We extended the description in the manuscript and added the underlying equation (L. 222-225):

  $E_{ph}$ is the phase-dependent MER of $SO_2$, $E_{SO2}$ is the mean MER based on the observed amount of $SO_2$ and the sum of the duration of all phases, $E_{ph}$ is the phase plume height (above the vent), and $H_T$=11571.2 m is the phase duration-weighted mean plume height derived as:

  $$H_T = \frac{\sum_{i=1}^{10} H_{ph,i} \cdot t_i}{\sum_{i=1}^{10} t_i}$$

- **Line   237: Please check the dates, aren't they 21 June and 22 June, instead?**

  No, the dates are correct. However, we added the UTC-times for clarification (L. 269):

  June 22, 0-23 UTC and June 23, 0-23 UTC

- **Line 239: (compare Fig. 3 top and bottom)**

  No, we here compare the horizontal spreading of ash and SO2.

- **Line 242: is the temporal evolution of ass loading on an hourly basis?**

  We only wrote output for the 3D field every full hour for the original version, although our time step is 60 s. More details are given in one of the comments

below.

- **Line 243: plateau? Is not a peak??**

We rephrased the sentence as (L. 275-277):

The Himawari-8 data reveals a steep increase of ash mass at 22 UTC on June 21 until a peak of 1.0 Tg is reached at 5 UTC on June 22 and the curve remains above 1.0 Tg for 5 hours. The maximum at 7 UTC (June 22) of 1.1 Tg is followed by a descent to 0.3–0.5 Tg.

- **Line 244: please start a new line when „Muser et al.…"**

Done (L. 278).

- **Figure 3: Please double check the dates**

The dates are correct here.

- **Line 246: please double check the timing, I guess it is 18 UTC of 21 June as the plot shows the hours since the June 21, 12UTC**

Yes, we corrected the date (L. 280).

- **Line 251: in the bracket add: in Figure 4**

Done (L. 284).

- **Line 254: here you refer to the effect of meteorological conditions. I guess it would be useful to add come vertical profiles of wind speed in correspondence of the vent.**

We found your comment very helpful to further explain plume dynamics and to highlight the advantage of FPlume instead of Mastin for dispersion forecasts. Therefore, we placed Fig. A2 to the appendix and we added the following description (L. 207-213):

The vertical profiles of the meteorological variables in Fig. A2a indicate increasing temperatures in most levels below 10 km during the long eruption phase between 9 and 14 h after simulation start. Additionally, the specific humidity increases by up to 1 g kg−1 in the lower 2 km (Fig. A2c). When warmer and moist air is entrained into the plume, the plume density reduces faster due to the lower ambient air density and the release of latent heat. This effect results in a higher positive buoyancy and a lower MER to reach a fixed height. In addition, the wind speed decreases in the lower 4 km between 9 and 14 h after simulation start, which reduces the plume bending and subsequently the MER needed to reach a fixed height.

- **Figure 4: Please be consistent with terminology: is this ash or fine ash? Are these hourly values? I'd much like to see the occurrence of pulses in these plots, to understand when the eruption dynamics affected the detection.**

  We corrected the title. Although our simulation time step is 60 s, the output of the 3-D fields (meteorological variables and tracers) of the simulations were written only every full hour and therefore the plot shows hourly values. We added the following sentence to the manuscript (L. 275):

  The temporal resolution of the data is 1 h.

- **Line 276: here you are suggesting that the fact that you are not describing the gravitational spreading occurring nearby the source is affecting your capability in reconstructing the SO2 forecast in its initial phase. Then please explain why this is not also impacting the quality of the ash forecast?**

  We realized that this sentence caused confusion. We removed it completely, as it is outside the scope of this paper.

- **Line 281: so from Figure 4 it turns out that the total amount of ash compares well with the satellite retrieval. However figure A2 shows a wider extent than what is retrieved. In this sense I'd have expected that this meant forecasted concentration lower than the observed and in this sense an Amplitude < 0 (as at Line 223-225 you say that when the model overestimates the parameter A is positive, and negative otherwise). But Figure 5 shows positive Amplitude for ash at all time intervals. Please explain.**

  The Amplitude indicates that the mean column loading in the whole domain considered. On the one hand, the higher amplitude values can be explained by the higher mean column loading within the modeled ash plume. On the other hand, we can see the impact of the zero values in the domain which cover more grid cells when the plume is more dispersed. We added to the manuscript (L. 317-320):

  The high amplitude value for ash between 12 and 36 h, despite the almost perfect agreement in the total mass in Fig. 4, also stems from the larger spread of the ash plume in the beginning. The reason is that the background values are considered zero and the Amplitude in the SAL analysis, unlike object-based structure and location values, is a domain-averaged quantity.

- **Line 285: here you say that S for ash in the beginning is negative. You justify the same results for SO2 saying that this is most likely due to the**

**ash thick presence that obscure the gas component in the retrieval. What might be affecting the negative S for ash?**

We removed this sentence because it is not visible in the data anymore after the re-run of the simulation.

- **Paragraph 3.3: please double check the dates you refer to…**

We double checked the dates and confirm that these are correct.

- **Figure 6: please clarify how the different heights are obtained. Here I see grey heights, are they coming from Fplume or satellite? Please specify. How are you getting the different heights for SO2 and ash? Are you maybe referring to the cloud height?? Please clarify.**

We added explanations of our methodology in the caption of Figure 3 as:

The gray bars indicate the duration and height of the 10 individual eruption phases. They are based on the analysis of the GOES-17 data, which serve as inputs for FPlume.

However, as stated above, we refer to the tracers originating from the volcanic emissions as the plume.

- **I guess the caption in Figure 7 and A3 partly clarifies it as it mention this „horizontally averaged vertical distribution of mass". Please spend more words in explaining the methodology, referring to Muser et al. 2020, is not enough. And use the words „plume" and „cloud" in a distinct manner.**

We added the description of our methodology (L. 329-332):

The plume top height in a) is defined as the maximum height of all grid cells in the plume that was separated from background mixing ratios as explained in Sect. 2.2.1. The average plume height in a) is the mean height weighted by the mass of all grid cells considered as inside the plume. The values in b) and c) were horizontally averaged over the whole detected plume,again excluding grid cells outside the plume. In b) and c), we picked June 23, 12 UTC, because it allows a direct comparison to Fig. 8 in M20, which only shows the ash plume top height.

Concerning the usage of 'plume' and 'cloud', please see the comments above.

- **Line 358: I think the statement about the „dense ash plume" hampering the validity of the comparison needs to be demonstrated. Please rephrase the conclusion.**

We agree with the reviewer, that the conclusion is too strong here, as we only speculated in the result. Thus, we rephrased it (L. 398):

However, we hypothesize that the validation of the simulated ash and SO2 dispersion was partially hampered by a dense ash plume in the beginning of the eruption and by overlapping water and ice clouds later on.

---

## Author Comment (AC3)

We thank Dr. Mingari for the constructive comments and suggestions, which helped us a lot to improve the manuscript.

We decided to re-run our simulations due to different comments in all three reviews. This leads to slightly different results, but does not change the main arguments of the paper. Please find our answers to your comments below (Reviewer's comments in **bold**, our replies in standard font, and modifications of the text in blue).
* * *
- **l. 51, p. 2: "They described the gravitational spreading of the umbrella cloud by the model of Costa et al. (2013). Collini et al. (2013) highlighted a good agreement in ash transport simulations with satellite observations for the Cordon Caulle eruption 2011 by a combined WRF/ARW-FALL3D forecast system." These introductory sentences seems out of context. Please better contextualize this discussion.**

  We rephrased this paragraph and removed details that are not relevant for our paper ( L. 49-53):

  Marti et al. (2017) overcame this issue by coupling the MMB-MONARCH-ASH transport model (Nonhydrostatic Multiscale Model on the B-grid – Multiscale Online Nonhydrostatic AtmospheRe CHemistry model – ASH)   with the 1D plume model FPlume, which calculates the MER and the mass distribution in the column online. Another example is the study by Collini et al. (2013), who combined the WRF/ARW forecast system with FALL-3D and highlighted a good agreement in ash transport simulations with satellite observations for the Cordon Caulle eruption 2011.

- **Eq. (1): The expressions for the vertical profile seem to be wrong in Marti et al. (2017) and also here. The Suzuki distribution should be normalized by the integral of S, instead of the maximum S. For discrete point sources, you should normalize using the sum of S requiring: E = \sum S\***

  We realized that the description here was misleading. We clarified it in the manuscript and added the following lines (L. 130-133):

  Eq. 1 explains the shape of the emission profile used here, which is also plotted in Fig. A1 in comparison with other profiles.To ensure the correct total ash mass emission and units when the particles are released into ICON-ART at discrete point sources in each model layer between the bottom and top height of the plume, we further normalized Eq. 1 by the integral of $S^*(z)$ (Rieger et al., 2015).

- **It seems that you distinguish between the terms "eruption phase" and "puff" in some parts of the manuscript, while sometimes are used as synonyms. In order to avoid confusion, it would probably be more convenient to unify the terminology and use only "eruption phase". In my opinion, "puff" is a bit ambiguous for a complex multi-phase eruption like Raikoke.**

  We replaced all ‚puffs' by ‚eruption phase' and additionally included the following explanation in L. 78-79:

  Throughout the paper, we define 'eruption phase' as one distinct time period in which the volcano was erupting.

- **What do you mean by "insensitive" here? Variations of 10% in column heights didn't affect MER estimations? Please clarify.**

  We added 'vent conditions' here (L. 160):

  The resulting MERs are insensitive to the input vent conditions (temperature, velocity, volatile fraction) in the range of 10%.

- **Figure 2 (title): "MER of very fine ash (<30 um)" → It should be "<32 um", right?**

  We corrected the title.

- **l. 217, p. 9: "The method assesses predefined objects based on a threshold value" You don't say how these objects were defined. What threshold value have you used?**

  We used a threshold of 0.2 g/m² and 2.5 g/m² for ash and SO2, respectively. We added the following paragraph (L. 262-265:):

  To define objects in the SAL analysis, we used a threshold of 0.2 g m-2 for modeled and observed ash, because this is the detection threshold for the Himawari-8 ash retrievals. For SO2, a threshold of 2.5 g m-2 for model and observations is used to remove background SO2 concentrations in Himawari-8 data. This was necessary, because we did not initialize the model with realistic background conditions and, therefore, can only compare the observed and modeled SO2 plume from the eruption.

- **l. 230, p. 9: Where do these gaps come from? The mean averaging you applied to fill gaps conserves the total mass? Or are you adding new mass with this approach?**

The raw Himawari-8 data column loadings are dense and without gaps in the native format. The mass evolution in Fig. 4 is calculated based directly on this native format (no mass changes). However, for the SAL analysis we needed to map the data on a regular lat-lon grid. The re-projected pixels are adjacent near the centre of the scan, pixels get larger towards the edges (latitudes>30°N and similarly for longitudes ±30° from the subsatellite point) and the separation between adjacent pixels also increases. In the region of the Raikoke volcano the scan angles and adjacent pixels are ~2 km apart (the nominal H8 resolution for IR). This leads to gaps on the lat-lon grid. There may be some mass changes but actually we only retrieve mass loading (not mass) so there are some implicit assumptions about the area of pixels etc which are more important. We added to the manuscript (L. 258-260):

These gaps in the satellite data arise during mapping from the native format onto a regular lat-lon grid as needed for the SAL analysis and are due to the increasing pixel sizes towards the edges of the retrieval domain.

- **l. 232, p. 9: Regridding mass loading to a coarser grid by a linear interpolation is not the best approach, as mass conservation is not ensured. Do you have an idea how much is the total mass difference induced by the interpolation method?**

We agree that linear interpolation is not the best approach. However, it does not makes much difference in the mass loading though as the field is smooth and the error is therefore negligible. An example is shown in the following figure for ash column loading in g/m² for two time averages (top: no interpolation; bottom: with interpolation).

[Figure]

Fig. 1: Himawari ash 6-h- averages for 18-24 h (left) and 24-30 h (right) after June 21, 12 UTC without filling the gaps in the plume that arise due to mapping on a regular latitude-longitude grid.

[Figure]

Fig. 2: Himawari ash 6-h- averages for 18-24 h (left) and 24-30 h (right) after June 21, 12 UTC with filling the gaps in the plume that arise due to mapping on a regular latitude-longitude grid.

- **l. 260, p. 10: "Thus, we conclude that the online treatment of plume development improves the ash loading prediction during the first hours and days after the eruption."This statement is not correct. It cannot be concluded from the results presented in Section 3.1 that the online treatment improves simulations. In fact, FPlume-norad outperforms Mastin-rad, and you cannot say that FPlume-rad is better than FPlume-norad. I think the only valid conclusion here is: "the FPlume experiments (ie, FPlume-rad and FPlume-norad) showed better agreement with observations"**

With online treatment, we refer to the experiments where MER are calculated within the simulation depending on the atmospheric state. Thus, the conclusion "that the online treatment of plume development improves the ash loading prediction during the first hours and days after the eruption" is valid. We clarified in L. 194-196:

The experiments FPlume-rad and FPlume-norad calculate the ESPs online within the simulation, whereas in Mastin-rad the ESPs are derived offline independent of the atmosphere and vent conditions.

- **l. 264, p. 11: SAL requires defining model and observation objects. For instance, you can identify clusters of grid cells with mass loading exceeding a given threshold. However, you haven't mentioned what threshold was used. This threshold can be defined based on the detection limit of the satellite retrievals. For example, Prata et al. (2021) assumed 0.2 g/m2 for volcanic ash. Otherwise, model-observation**

**comparisons wouldn't be fair. Please clarify what thresholds you have assumed for ash and SO2.**

Done (see answer to comment above).

- **l. 282, p. 12: Results in Fig. A2 are really good. I think it would be worth including Fig. A2 in the main body of the paper, probably replacing Fig. 3.**

  We produce these two figures (3 and A2) for different reasons: Fig. 3 should give the reader a short overview on the location and the dispersion of the cloud without showing too many details. Figure A2 should give supporting information to understand the values of the SAL analysis and is much too large for the main text. We therefore leave the order of the figures as it is.

- **l. 295, p. 12: Why are you defining two threshold for volcanic ash instead of using a single threshold for total ash? Why didn't you define a threshold for the giant mode? What threshold have you used for ash in Fig. 6?**

  We included a subsection (Sect. 2.2.1) in the Methods part describing the choice and need of the threshold values.

- **l. 296, p. 12: I see no reason to remove those 'steps'. Why top height is greater than zero before the eruption starting time in Fig. 6? This has to do with the smoothing? Since data was smoothed, you should at least indicate the vertical resolution of the model. Is it comparable to the differences found in top height between rad and no-rad experiments?**

  We attached the figure without the smoothing and with the 'steps' here, but we decided to keep the figure with the smoothed lines in the manuscript. The differences in the plume top height between the smoothed data and not-smoothed data are only small. We agree that the top height >0 km before the eruption start, which results from the smoothing, is misleading. We therefore adjusted the strength of smoothing and added the following sentence in L. 335-336:

  The difference in height between FPlume-rad and FPlume-norad remains, regardless of the use of this filter. However, the increasing plume height already starting before the beginning of the eruption is a result of the filtering.

[Figure]

Fig. 3: a) Temporal evolution of the SO2 (red) and ash (yellow) plume top height and mass-averaged height for the FPlume-rad (solid) and FPlume-norad (dashed) experiment. The grey bars indicate the duration and height of the 10 individual eruption phases; b): vertical profile of the temperature difference between FPlume-rad and FPlume-norad in the ash plume 48 h after the start of the simulation. c): vertical SO2 (red) and ash (yellow) profile averaged over the plume after 48 h for the FPlume-rad (solid) and FPlume-norad (dashed) experiment.

- **Figure 6: How is the mass averaged height computed? Is a vertical average? In this case, the average is weighted by mass concentration or by mixing ratios? Or is a horizontal average? In this case, the average is weighted by mass loading (in g/m2). Are vertical profiles also averages? Or are they computed at specific locations? How this average is performed? Do exclude grid cells without ash/SO2 from the average?**

We included the following information (L. 329-332):

The plume top height in a) is defined as the maximum height of all grid cells in the plume that were separated from background mixing ratios as explained in Sect. 2.2.1. The average plume height in a) is the mean height weighted by the mass of all grid cells considered as inside the plume. The values in b) and c) were horizontally averaged over the whole detected plume, again excluding grid cells outside the plume.

- **l. 306, p. 13: "in the FPlume-norad case still shows the overlap of the different phase dependent profiles" Where is shown this overlap? Please clarify.**

We removed this sentence, as it does not make sense here anymore.

- **l. 318, p. 14: "the vertical distribution of the total ash mass"
What do you mean by "total mass"? Is the total mass within a model layer?**

We used a wrong title in the plot and also here in the text (see next comment). We corrected this to mass concentration in kg/m3.

- **Figure 7(c): "SO2 mass loading in kg" -> mass loading should be in units of g/m2 as in Fig. 3. Why are you showing "mass" in (a) and "mass loading" in (c)? What is the difference?**

Here, the title was wrong. In the first manuscript, both plots showed mass loading in g/m2 for the individual grid cells. However, we changed the quantities to concentrations in kg/m3 as this is independent of the height of the grid cell and easier to understand. Nevertheless, the argumentation in the text is not affected by this.

- **Equation (3): Obtaining the median radius for a multimodal log-normal distribution is not a trivial problem. Are you sure the median radius is given by such a simple formula? Or this expression only defines a "characteristic radius"?**

We agree with the reviewer and used the word ‚characteristic radius' instead throughout the text.

- **Figure A1(left) is not relevant.**

We provided this figure not only to explain our profile, but also for others as a reference and to reproduce our settings without further complications. Thus, we leave this figure as it is.

**Technical corrections and minor comments:**

- **l. 126, p. 5: Remove parentheses from Marti et al. (2017)**

Done.

- **l. 130, p. 5: Correct citing format in Rose and Durant (2009)**

Done.

- **l. 143, p. 5: It is necessary a reference for the Mastin empirical relationship**

Done.

- **l. 155, p. 6: Specify: "ellipsoid" -> "Earth ellipsoid"**

  Done.

- **l. 166, p. 6: What do you mean by "marine Sc layer"?**

  Done.

- **l. 174, p. 7: "The Raikoke eruption 2019" -> "The  2019 Raikoke eruption"**

  Done.

- **l. 204, p. 8: "Himawari-8 Ash and SO2" -> "Himawari-8 Ash and SO2 retrievals"**

  Done.

- **l. 212, p. 8: "The ash retrievals were corrected (...) completely cloud covered". Please clarify this sentence.**

  We extended the sentence like this (L. 137):

  The ash retrievals were corrected by a mask that accounts pixels that contain meteorological clouds but which were classified as completely cloud covered.

- **l. 330, p. 15: "Fig. 7 also shows" -> "Fig. 7d also shows"**

  Done.

- **l. 331, p. 15: "the radius is higher on average because" ->"the radius is higher on average according to the FPlume-rad experiment because"**

  Done.

- **Figure 7 (caption): "and SO2 mass (b)" -> "and SO2 mass (c)".**

  Done.

---

## Author Response (AR2)

We thank Peter Haynes for the comments and suggestions on our manuscript. Please find our answers to the comments below (Editors's comments in **bold**, our replies in standard font, and modifications of the text in blue).

**I have received one referee report on your revised paper -- that referee recommends that the paper is very close to being suitable for publication but questions your use of terminology -- in particular the use of 'plume' rather than 'cloud' which the referee believes will be very confusing to those in volcanic science community. The referee suggests a way around this -- to use the term 'volcanic cloud' or 'volcanic ash cloud'. Please consider using that -- or something similar -- and provide justification if you really strongly wish to use 'plume' -- but in that case I suggest that you insert a sentence that makes it explicit that you are using 'plume' to describe the passively advected ash/aerosol rather than the strong upward flow emerging directly from the volcano.**

The focus of our paper is on the modeling of volcanic ash dispersion in the atmosphere. We therefore still prefer the term 'plume' instead of 'cloud' which is more common in the field of volcanology. However, we added the following sentence to the manuscript for clarification (L.64-66):

Throughout this paper, we use the term 'plume' or 'volcanic plume' to describe the part of the volcanic material dispersed in the atmosphere as commonly used in the meteorology instead of 'volcanic cloud' to maintain a clear distinction from the meteorological clouds.

**With regard to the comments on the original paper by Referees 1 and 3 -- my impression is that you have provided thorough responses those but in some cases -- particularly with regard to comments of Referee 1 -- you have not made a change to the text. That is fine in principle -- there is no rule that says that a change to the text has to be made, and in the ACP system your responses to the referees will be publicly visible -- but I wonder with respect to some of the comments whether a change to the text would be helpful to the reader. (The referee is requesting more information on a certain point -- you have provided more information in the response -- but a reader might well have a similar question and benefit if a bit more information -- something brief -- is provided in the text.) Please can you look again at the responses and think about whether a sentence or two added to the text would help a reader of the paper, particularly if they were thinking in the same way as the referee.**

We added the following:

- L. 137-143
  We only used the MER from FPlume and calculated the very fine ash fraction and emission profile independently due to two main reasons: (1) based on offline analysis we figured out that the mass profiles for the predefined bin sizes strongly depends on the assumption of initial total grain size distribution (TGSD). As information on the TGSD is often lacking, using FPlume mass profiles lead to a less generic approach and large uncertainties. (2) the definition of ash modes in ICON-ART is only relevant for long range transport in the atmosphere and differs from TGSD at the vent. Thus, we would have to convert the FPlume size bins into ICON-ART mode which requires several assumptions and increases the uncertainty of the emissions.
- L. 308-313:
  We do not discuss the SAL values for the FPlume-norad and the Mastin-rad case here, because FPlume-norad only shows very small differences to the FPlume-rad case in all SAL

values and Mastin-rad only changes the amplitude value as only MER is higher compared to Fplume-rad. Based on the analysis of hourly to daily mean values, we conclude that 6-hour averages provide a reasonable compromise between both reproducing the details and reducing the amount of missing values and noise.